# Spatio-Temporal Changes in Forest Area and Its Ecosystem Service Value in Ganzi Prefecture, China, in the Period 1997–2017

**Yanru Wang** [1,2,3], **Qingquan Li** [1,*], **Jijin Geng** [2,*], **Xiaojuan Bie** [3], **Peihao Peng** [3] **and Guofeng Wu** [1]

1   School of Architecture and Urban Planning, Shenzhen University, Shenzhen 518060, China;
    wangyr@szu.edu.cn (Y.W.); guofeng.wu@szu.edu.cn (G.W.)
2   Shenzhen Marine Development & Promotion Center, Shenzhen 518034, China
3   College of Earth Sciences, Chengdu University of Technology, Chengdu 610059, China;
    biexiaojuan06@cdut.cn (X.B.); pengpeihao@cdut.edu.cn (P.P.)
*   Correspondence: liqq@szu.edu.cn (Q.L.); gengjijin@szpgzx.com (J.G.)

**Abstract:** It is possible to manage the forest ecosystem and promote sustainable development by keeping track of spatio-temporal fluctuation in the forest area and its ecosystem service value (ESV). The forest ecology of Ganzi Tibetan Autonomous Prefecture (Ganzi Prefecture), which is located in the northern Hengduan Mountains region, i.e., China's most important ecological functional area, has seen significant alteration during the past 20 years. However, little is known about how the forest and its ESV evolve. We obtained data regarding Ganzi Prefecture's forests using visual interpretation of remote sensing images derived from 1997, 2007, and 2017, and we evaluated the spatial–temporal changes in the forest ESV from 1997 to 2017 using global value coefficients and adjusted local value coefficients. The results revealed that (1) from 1997 to 2017, the forest area of Ganzi Prefecture increased by 6729.95 km², and the forest growth rate was 336.50 km²/a, while (2) from 1997 to 2017, the forest ESV in Ganzi Prefecture experienced an overall increase of $257.59 \times 10^8$ yuan. The primary driver of the forest ESV increase was the implementation of forestry ecological engineering and protection policies. (3) Finally, the spatial distribution of the forest ESV revealed that the forest ESV density increased during this period, with the most significant increase occurring in Yajiang. The forest ESV was scattered with the highest density in Yajiang and the lowest density in Shiqu. This study emphasizes how crucial forest ecosystems are to Ganzi Prefecture's mechanisms for maintaining life. It provided a scientific basis for the sustainable management of the forest ecosystem in the Hengduan Mountains.

**Keywords:** Hengduan Mountains; forest ecosystem service value; forestry protection project; forestry policy and management





## 1. Introduction

In recent years, there has been a lot of interest in quantifying ecosystem services and their intrinsic value; experts and academics concur that "ecosystem services" should be considered when making decisions about how to manage natural resources [1–3]. Ecosystem services are direct or indirect benefits that humans derive from ecosystems. They include a wide range of goods and services that contribute to the global economy's value, with key indicators including delivering services, providing cultural services, supporting services, and regulating services, which are all essential for human well-being, economic growth, and long-term human development [4]. ESV is a monetized statement of ecosystem services and a tool used to examine the inter-relationships between human economies and societies and natural ecosystems. Accurate global and regional ESV evaluations have substantially increased human understanding of natural ecosystems [5] and supplied critical information used to create good ecological preservation programs [6].

In 1997, Costanza conducted the first quantitative analysis of the worldwide ESV, providing the groundwork for ESV research, and proposed a method of assessing the ESV that has received significant scholarly attention [7–9]. However, this approach has certain limitations, including the following issues: (1) the determination of value coefficients is rather subjective, and (2) whether the value coefficients are accurate for the actual situation in the study area will directly affect the accuracy of the assessment. To solve this problem, Xie et al. [10,11] created a revised coefficient scale that was suitable for China and based on the features of terrestrial ecosystems. Chinese researchers frequently evaluate ESV using their enhanced methodology, which has the advantages of requiring low data-gathering expenses and having an easy evaluation procedure [12,13].

As an important component of terrestrial ecosystems, forests provide a wide range of ecosystem services to humans [14–19]. The ecosystem services provided by forests are the basis for sustaining social, ecological, and economic development [13]; essential to human existence and development [20]; and intimately linked to human well-being and national ecological security. Understanding the dynamic shifts in the distribution of forest resources and measuring forest ESVs provides solid support for resource management and conservation decision-making [21,22], and it helps decision makers to achieve the goal of precision management. The spatial and temporal dynamics of forest resource distribution and their ecosystem service values (ESVs) are topics of concern for the international community today, and policymakers are paying ever-more attention to them. Therefore, in order to address the concerns regarding the vulnerability, resilience, sustainability, and human-linked survival of forests, it is crucial to identify the spatial and temporal dynamics of forest resources and their ESVs, whether globally or in a specific location.

At present, spatial information technology represented by remote sensing (RS) and geographic information system (GIS) provides a convenient and effective method of obtaining spatial and temporal dynamic changes in forest resources and their ESVs in large areas [23–27]. Remote Sensing, as a new type of earth observation technology, has strong macroscopic and diverse information acquisition methods, has rapid and real-time advantages, and is an essential tool in terms of acquiring various types of spatial information [28–32]. GIS is an important tool in terms of the spatial visualization of information, and the emergence and growth of the two factors have brought about a qualitative leap in the extraction, monitoring, analysis, and management of features; expanded the research content and paradigms of various disciplines; provided a powerful detection tool that provides rapid access to spatio-temporal dynamics information about forests; and provided an important technical means to achieve the effective mastery of spatio-temporal dynamic information about forests and their ESVs, as well as for the management and protection of forests [33–36].

As the northern part of the Hengduan Mountains, Ganzi Prefecture is the most significant ecological functional region in China [37–40], and it is a biodiversity conservation hotspot of global concern and a critical water-conserving zone in the upper Yangtze River, China. Ganzi has a critical position in China's forest reserve strategy and is the central part of the second largest natural forest area in China, i.e., the Southwest Forest Area, the forest types and distribution of which are typical and representative of the southeastern edge of the Tibetan Plateau, as well as crucial for regulating China's climate, maintaining the balance of carbon dioxide in the atmosphere, conserving water, stopping soil erosion, and protecting the country's ecological environment [39]. Before 1998, the forest ecosystem in this area was severely damaged. To restore the forest habitat in this region, the state implemented specific forestry ecological projects and protection policies in Ganzi Prefecture. The distribution and service functions of the forest ecosystem in Ganzi Prefecture have undergone significant modifications in the last 20 years. However, the spatial and temporal changes in the forest area and forest ESV in Ganzi Prefecture are not yet fully understood, and information is significantly lacking for the Hengduan Mountains. Therefore, this study is urgently needed.

Based on the above research gaps, this study used remote sensing, GIS spatial analysis techniques, and revised equivalent factor methods to quantitatively analyze the spatial and temporal changes in the forest area and forest ESV in Ganzi Prefecture from 1997 to 2017. There are two specific objectives:

(1) Identifying the spatial and temporal variations in the forest area from 1997 to 2017;
(2) Revealing the temporal and spatial variations in forest ESV and making suggestions regarding forest development and management.

This study is essential for the rational protection and management of the forest ecosystem in Ganzi Prefecture, promoting the sustainable development of forest ecosystem services. These findings serve as a scientific guide for developing methods to protect forests in the Hengduan Mountains and other regions.

## 2. Materials and Methods

### 2.1. Study Area

Ganzi Prefecture (97°22′ E to 102°29′ E, 27°58′ N to 34°20′ N) is situated in the southwest of China's Sichuan Province, which is, in turn, located on the southeastern edge of the Qinghai–Tibet Plateau (Figure 1). It is an immensely ecologically significant location in China and lies in the Hengduan Mountains region [37–40]. It has jurisdiction over 1 county-level city and 17 counties, namely Yajiang, Luding, Danba, Daofu, Baiyu, Ganzi, Jiulong, Huhuo, Litang, Xinlong, Dege, Shiqu, Sedar, Batang, Xiangcheng, Dacheng, Derong, and Kangding, with the state capital being Kangding. It is 663 km long from north to south and 490 km wide from east to west, and it has a total area of $15.3 \times 10^4$ km², making up 31.76% of the total area of Sichuan Province. The region's topography belongs to the alpine plateau region of the northern Hengduan Mountains in western Sichuan. It is a crucial component of the Qinghai–Tibet Plateau. The terrain is high in the northwest and low in the southeast. It is broadly divided into three types of terrain: hilly plateau, mountainous plain, and high mountain valley, and it has an average altitude of over 3500 m, with Mount Gongga being the highest peak at 7556 m. The region's climate is highland monsoonal, with over 2000 h of annual sunshine and low precipitation, which ranges roughly from 320–800 mm per year. There are many rivers in the region, including the Dadu, Yalong, and Jinsha Rivers, which are all significant tributaries of the Yangtze River's upper reaches.

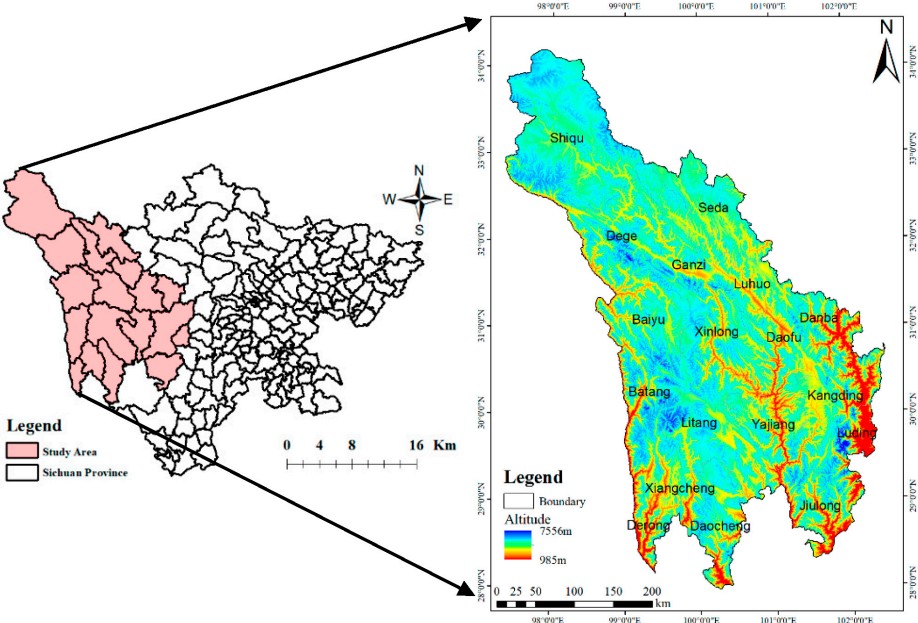

**Figure 1.** Location of Ganzi Prefecture.

## 2.2. Remote Sensing Data Processing and Forest Information Extraction

In this study, forest information from 1997, 2007, and 2017 was extracted using Landsat 5TM (Thematic Mapper, TM) remote sensing images in 1997 and 2007, as well as Landsat 8OLI (Operational Land Imager, OLI) remote sensing images in 2017. The spatial and spectral information of Landsat 5TM and Landsat 8OLI remote sensing data is shown in Tables 1 and 2, respectively, and the remote sensing images were 1 June 2019 downloaded from the Geospatial Data Cloud (http://www.gscloud.cn).

**Table 1.** Spatial and spectral information of Landsat 5TM remote sensing data.

| Band Name | Spectral Range (μm) | Resolution (m) | Main Application |
|---|---|---|---|
| B1-Blue | 0.45–0.52 | 30 | Water penetration and distinguishing soil and vegetation |
| B2-Green | 0.52–0.60 | 30 | Distinguishing vegetation |
| B3-Red | 0.63–0.69 | 30 | Observation of roads, bare soil, vegetation types, etc. |
| B4-NIR | 0.76–0.90 | 30 | Biomass estimation |
| B5-SWIR | 1.55–1.75 | 30 | Distinguishing roads, bare soil, water, vegetation types, etc. |
| B6-LWIR | 10.40–12.5 | 120 | Sensing targets emitting thermal radiation |
| B7-SWIR | 2.08–2.35 | 30 | Recognizing rocks and minerals, as well as vegetation cover and moist soil. |

**Table 2.** Spatial and spectral information of Landsat 8OLI remote sensing data.

| Band Name | Spectral Range (μm) | Resolution (m) | Main Application |
|---|---|---|---|
| B1-Coastal | 0.43–0.45 | 30 | Used to observe the coastal zone |
| B2-Blue | 0.45–0.51 | 30 | Used for water penetration and to distinguish soil and vegetation |
| B3-Green | 0.53–0.59 | 30 | Used to distinguish vegetation |
| B4-Red | 0.64–0.67 | 30 | Used to observe of roads, bare soil, vegetation types, etc. |
| B5-NIR | 0.85–0.88 | 30 | Used to estimate biomass, discriminate wet soil, etc. |
| B6-SWIR1 | 1.57–1.67 | 30 | Used to distinguish roads, bare soil, water, vegetation types, etc. |
| B7-SWIR2 | 2.11–2.29 | 30 | Used to recognize rocks and minerals, as well as vegetation cover and moist soil |
| B8-Pan | 0.50–0.68 | 15 | Black and white images at 15-m resolution used to perform enhanced resolution |
| B9-Cirrus | 1.36–1.38 | 30 | Used to perform cloud detection, cloud removal, and other applications |
| B10-TIRS1 | 10.6–11.19 | 100 | Targets for sensing thermal radiation |
| B11-TIRS2 | 11.5–12.51 | 100 | Targets for sensing thermal radiation |

According to the growth pattern of forest vegetation, summer is the period with the highest forest vegetation coverage and the most vigorous growth. Therefore, we tried to select remote sensing images taken from June to October with less cloud cover (less than 10%) in this study. For the images that could not meet the requirements, the images collected in the same period were replaced by the images of the neighboring years, ensuring that the images used for the three time periods cover the whole territory of Ganzi Prefecture. Finally, 46 landscape images were selected.

The methodology and process used in this study is shown in Figure 2. The specific remote sensing image processing and forest information extraction process used was defined as follows:

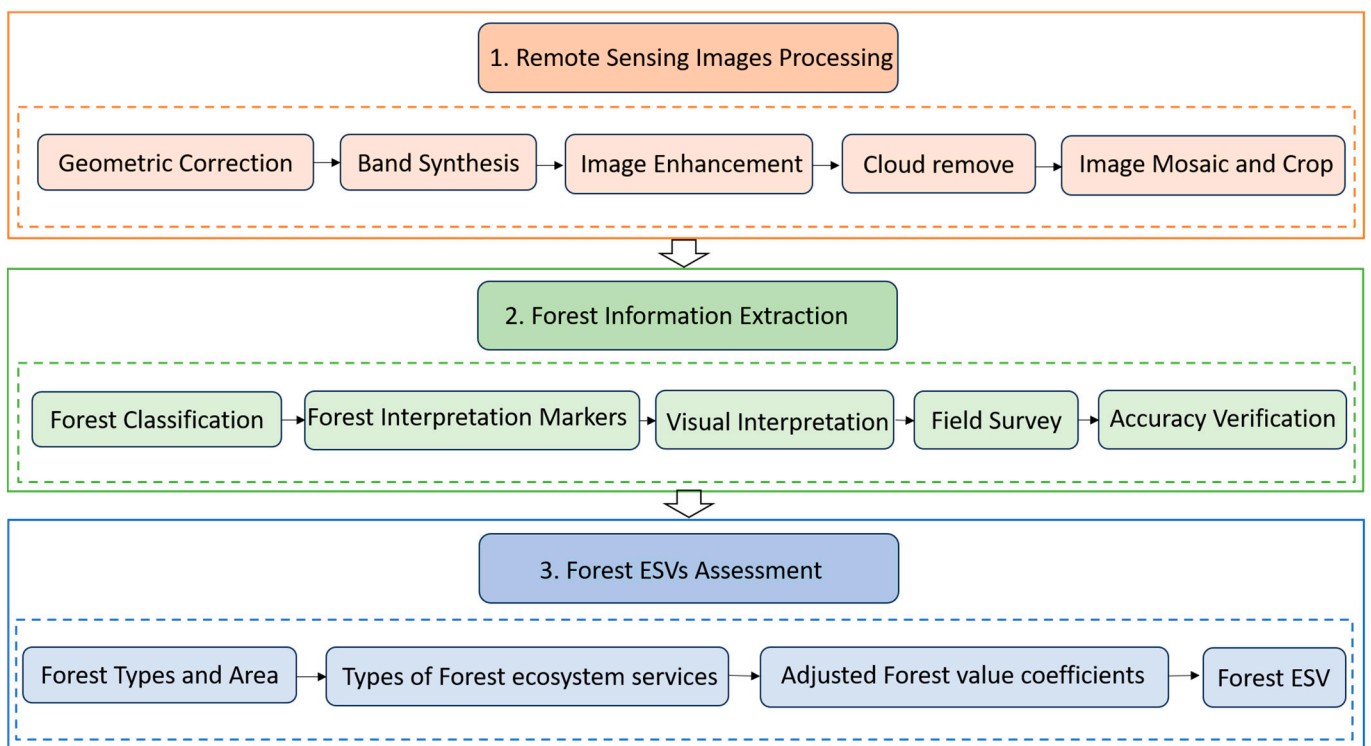

**Figure 2.** The approach and workflow used in this study.

Firstly, the remote sensing images were subjected to data pre-processing, such as performing atmospheric correction, geometric correction, band synthesis, cloud removal, shadow processing, image enhancement, image mosaic, and cropping. As the standard false color remote sensing images were characterized by highlighting vegetation information, which could be used to perform vegetation monitoring, this study adopted the standard false color synthesized remote sensing images to perform forest information extraction. Among these images, Landsat 5TM images were processed with 4, 3, and 2 bands and synthesized, and Landsat 8OLI images were processed with 5, 4, and 3 bands and synthesized.

Then, based on the Chinese terrestrial ecosystem classification system of the Resource and Environment Database of the Chinese Academy of Sciences, the Geography and Resources of Trees in Ganzi Prefecture, the National 1:1 Million Vegetation Type Map, the Vegetation of Sichuan, the Forestry Journal of Ganzi Tibetan Autonomous Prefecture, the historical information of forests in Ganzi Prefecture, and field forest survey information, in addition to the features of remote sensing images (spectrum, texture, etc.), and taking into account the aspects of forests located in the Hengduan Mountains of Ganzi Prefecture and natural and economic development characteristics, a forest classification system suitable for the remote sensing monitoring of forests in Ganzi Prefecture was established.

Since visual interpretation is a common method used to extract forest information using remote sensing images, it has the advantage of high accuracy [41–43]. Thus, this study combined the natural conditions of Ganzi Prefecture, forest characteristics, ground truth data regarding Ganzi forests obtained through Global Positioning System (GPS) field surveys, etc., as well as using existing forest data, Landsat images, and Google Earth high-resolution images as reference data [30], the visual interpretation markers of forest ecosystems were established, relying on the morphological, texture, and tonal features of different forest types identified via the false color images.

Finally, the pre-processed remote sensing images and the established forest interpretation marker were employed to perform visual interpretation of the forests of Ganzi Prefecture, and at the same time, a large number of field investigations and verifications of the interpretation results were carried out by combining the natural conditions of Ganzi Prefecture, the forest characteristics, the high-resolution remote sensing images of Google Earth, and the references to the auxiliary materials, such as the historical field surveys, etc., finally obtaining 30-m spatial resolution forests distribution maps from 1997 to 2017 (Figure 3) after a series of supplementary interpretations and corrections. To further confirm the accuracy of the forest data, this study used the confusion matrix and Kappa coefficient method to verify the accuracy of the forest information extracted via visual interpretation in the period 1997–2017, and the results showed that the overall classification accuracy and Kappa coefficient of the three forests distribution maps were all greater than 90%, which indicated that the extracted forest information was reliable, as well as that the classification accuracies fulfilled the standards required for forest ecosystem research.

*2.3. Forest ESV Assessment*

After extracting forest information, the next step was to perform a forest ESV assessment (Figure 2). According to the historical literature, i.e., the equivalent coefficients of ESV mentioned by Costanza and Xie [7–11], as well as the "Ecosystem Service Value Equivalent Scale for China's Terrestrial Ecosystem" by Xie Gaodi [10,11,44], the forest ecosystem includes 11 different types of ecosystem service functions. This study improved the forest ESV equivalent in Ganzi Prefecture, and the results are shown in Table 3.

**Table 3.** The Ganzi Prefecture's forest ESV equivalent coefficients.

| Forest Ecosystem Service | Needle-Leaf Forest | Broad-Leaf Forest | Mixed Forest |
|---|---|---|---|
| Food production | 0.22 | 0.29 | 0.31 |
| Raw material | 0.52 | 0.66 | 0.71 |
| Water supply | 0.27 | 0.34 | 0.37 |
| Gas regulation | 1.70 | 2.17 | 2.35 |
| Climate regulation | 5.07 | 6.5 | 7.07 |
| Purify environment | 1.49 | 1.93 | 1.99 |
| Hydrological regulation | 3.34 | 4.74 | 3.51 |
| Soil formation | 2.06 | 2.65 | 2.86 |
| Nutrient cycle | 0.16 | 0.2 | 0.22 |
| Biodiversity | 1.88 | 2.41 | 2.6 |
| Aesthetic landscape | 0.82 | 1.06 | 1.14 |
| Total | 17.53 | 22.95 | 23.13 |

According to Xie Gaodi et al., who used the net profit from grain production per square meter of farmland as a standard equivalent ecology factor, an ecosystem's value equivalent in China in 2010 was 3406.50 yuan/hm$^2$ [7,11,45]. This approach was based on the national grain per unit area yield and the average yearly production of grain per unit area in Ganzi Prefecture from 1997 to 2017 (4973 kg/hm$^2$ and 3019.89 kg/hm$^2$, respectively) [11]. Finally, the forest ESV of the unit equal factor in the study area was determined to be 2068.62 yuan/hm$^2$a, and the agricultural ecosystem service equivalent value coefficient in the research region was corrected to 0.61. The specific formulas used to calculate the forest ESV were as follows:

$$VC_i = \sum_{j=0}^{n} EC_j \times E_a \tag{1}$$

$$ESV = \sum_{i=0}^{n} A_i \times VC_i \tag{2}$$

*ESV* is the value of forest ecosystem services, *i* is a forest type, *j* is a forest ecosystem service type, $A_i$ is the area of a class *i* forest type (hm$^2$), $VC_i$ is the *ESV* per unit area of a class *i* forest type (yuan/hm$^2$a), $EC_j$ is the value equivalent of item *j* ecosystem services

of a certain type of forest, and $E_a$ is the equivalent factor of 2068.62 (yuan/hm$^2$a) for the economic value of a unit ecosystem service.

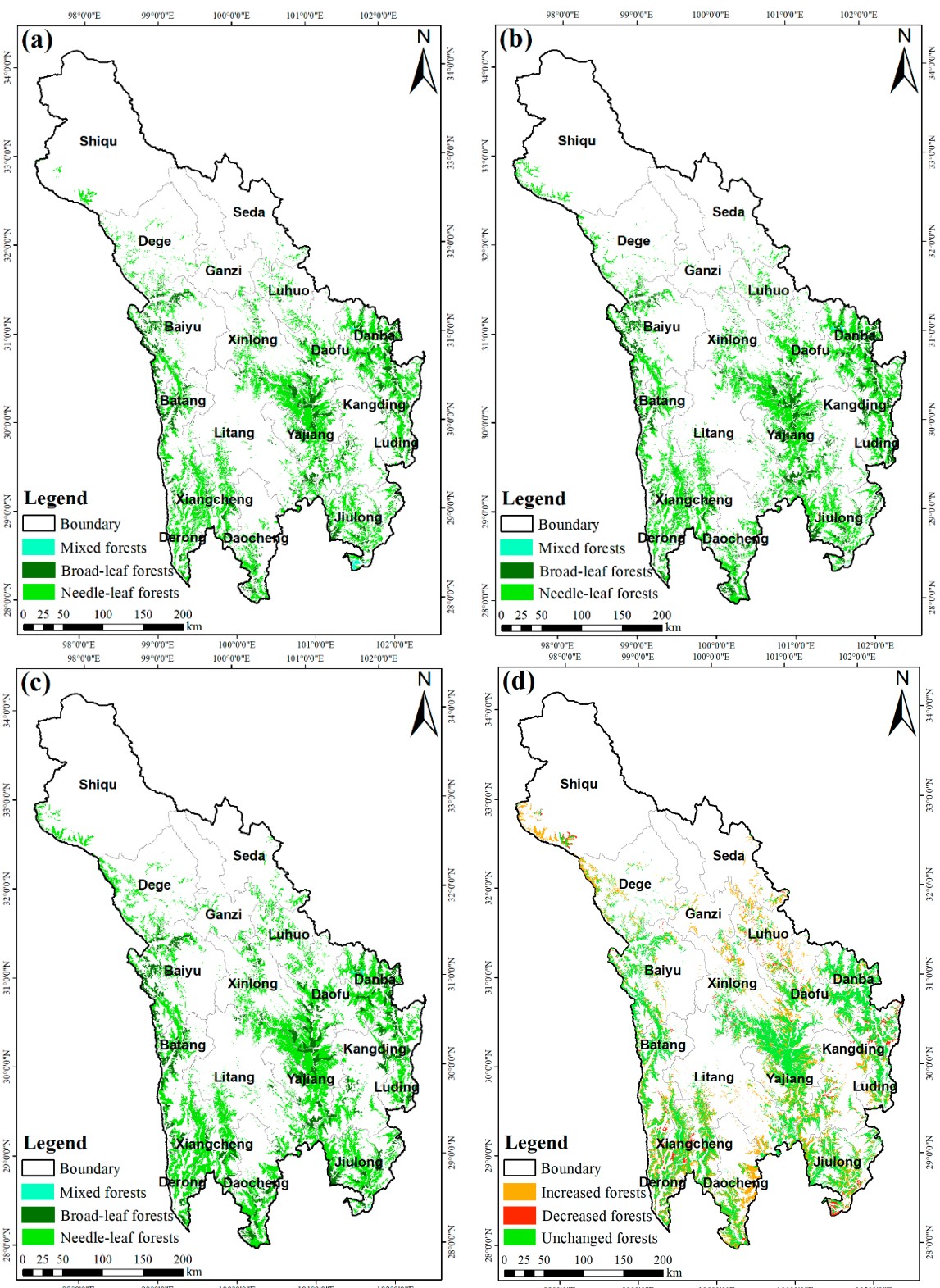

**Figure 3.** Spatial distribution and change maps of forests in Ganzi Prefecture from 1997 to 2017: (**a**) 1997, (**b**) 2007, (**c**) 2017, (**d**) changes in forests, and the period 1997–2017.

### 3. Results

*3.1. Changes in Forest Area*

The dynamic change in the forest area in Ganzi Prefecture from 1997 to 2017 is shown in Figure 3 and Table 4. From 1997 to 2017, the total forest area of Ganzi Prefecture showed an initially slow and then strong increasing trend, with a total increase of 6729.95 km$^2$ and an overall rate of change of 336.50 km$^2$/a. This result mainly occurred due to the fact that after 1997, China initiated and implemented a series of major forestry ecological projects and protection policies, such as the "natural forest protection project and ban on logging in natural forests", the "returning farmland to forests and grassland" project, and the "protection of wild fauna and flora and the construction of nature reserves" project, which led to an increase in the forests in the region. Among these changes, the degree of forest change during the period 1997–2007 was relatively gentle, recording a total increase of 1160.99 km$^2$ and a rate of change of 116.10 km$^2$/a, while the degree of forest change during the period 2007–2017 was more drastic, recording a total increase of 5568.96 km$^2$ and a rate of change of 556.90 km$^2$/a.

**Table 4.** Changes in the areas of different types of forests in Ganzi Prefecture from 1997 to 2017.

| Forest | Area (km$^2$) | | | Changes Rate (km$^2$/a) | | |
|---|---|---|---|---|---|---|
| | 1997 | 2007 | 2017 | 1997–2007 | 2007–2017 | 1997–2017 |
| Needle-leaf forests | 17,321.51 | 18,221.99 | 22,841.89 | 90.05↑ | 461.99↑ | 276.02↑ |
| Broad-leaf forests | 4639.63 | 4945.93 | 5887.61 | 30.63↑ | 94.17↑ | 62.4↑ |
| Mixed forests | 83.52 | 37.73 | 45.12 | −4.58↓ | 0.74↑ | −1.92↓ |
| Total | 22,044.66 | 23,205.65 | 28,774.61 | 116.1↑ | 556.9↑ | 336.5↑ |

Note: "↑" means increase, and "↓" means decrease.

In terms of the composition of forest types, the forests in Ganzi Prefecture were mainly composed of needle-leaf and broad-leaf forests, with the composition law being needle-leaf forests > broad-leaf forests > mixed forests, of which the areas of needle-leaf forests in Ganzi Prefecture in 1997, 2007, and 2017 were 17,320.80 km$^2$, 18,221.25 km$^2$, and 22,842.48 km$^2$, respectively, accounting for more than 78% of the total area of forests in Ganzi Prefecture. The areas of broad-leaf forests in 1997, 2007, and 2017 were 4640.34 km$^2$, 4946.67 km$^2$, and 5887.02 km$^2$, respectively, accounting for more than 20% of the total forest area, while the areas of mixed forests were lower. In terms of forest dynamics, both needle-leaf and broad-leaf forests in Ganzi showed an increasing trend from 1997 to 2017, recording rates of increase of 276.08 km$^2$/a and 62.33 km$^2$/a, respectively, while mixed forests showed a decreasing trend, recording a rate of decrease of −1.92 km$^2$/a.

In terms of the spatial dynamics of forests, it was found by overlaying the data regarding the spatial distribution of forests in Ganzi Prefecture in 1997 and 2017 that forests have spatially increased in some places and remained unchanged in others, while forests have decreased in other areas (Figure 3). Forests that were geographically spatially diminished typically degraded to scrub or grassland over time. Deforestation and natural disasters were the main causes, and deforestation was primarily brought on by human factors. Forestry was crucial to Ganzi Prefecture's economy at the time, and as a result of the long-standing implementation of the policy of heavy extraction and light cultivation, as well as heavy deforestation and light afforestation of the forests in Ganzi Prefecture, those forests suffered significant damage. Moreover, the forests in Ganzi Prefecture were primarily made up of natural and primitive forests with delicate ecosystems, and the recovery of such forests after the destruction was extremely sluggish. Natural forests were dominated by cold temperature zone *fir* and *spruce* trees, which had low growth rates and long growth cycles, and the effect of artificial regeneration was poor. In addition, forest fires occasionally broke out for natural or artificial reasons, and the rate of forest harvesting and burning was much higher than the rate of forest regeneration. The forest gradually degraded into thickets or meadows, causing the forests to spatially shrink.

The areas in which forests spatially increased were mainly located along the Jinsha, Yalong, and Dadu Rivers, replacing land-use types such as barren hills and wastelands, open forests, general shrublands, abandoned mines, logging tracks, and fire tracks. The primary reason for the spatial increase in and constancy of forests was the implementation of national forestry ecological projects and protection policies. Since 1998, China has carried out a number of significant forestry ecological protection projects and protection policies in Ganzi Prefecture, such as the "natural forest protection project and natural forest logging ban, "forest harvesting quota management," and other initiatives. As a result, the number of forests in Ganzi Prefecture has steadily increased through initiatives like the closure of mountains and forests, fly seeding and afforestation, and plantation afforestation, as well as other methods.

As demonstrated in Figures 3 and 4, as well as Table 5, the forests in Ganzi Prefecture showed a spatial distribution pattern of more forest in the south and fewer forests in the north, and there were obvious differences between the spatial distribution of forests in each county, occurring mainly due to the fact that Ganzi Prefecture was located in the southeastern edge of the Qinghai–Tibetan Plateau, which has a large undulating topography, diverse climatic conditions, and significant geographic variations. The distribution of forests was also affected by a wide range of factors, such as topography, climate, soil, and human activities. In 1997, 2007, and 2017, the counties with the largest forest areas were Yajiang, which had forest areas of 2970.32 km$^2$, 3084.52 km$^2$, and 3678.01 km$^2$, respectively, followed by Kangding, which had areas of 2194.07 km$^2$, 2446.18 km$^2$, and 2542.86 km$^2$, respectively. Jiulong ranked third, recording areas of 2011.38 km$^2$, 2130.04 km$^2$, and 2551.53 km$^2$, respectively, which indicates that the above locations were suitable for forest distribution. However, the county with the smallest forest area was Ganzi, recording areas of only 59.99 km$^2$, 46.13 km$^2$, and 198.56 km$^2$, respectively; Seda and Shiqu also have fewer forest areas, and the main reason for this issue was the high altitudes and low temperatures of Ganzi, Seda, and Shiqu, which limit the distribution of forests.

**Table 5.** Changes in forest area in different counties from 1997 to 2017.

| Forest | Area (km$^2$) | | | Changes Rate (km$^2$/a) | | |
|---|---|---|---|---|---|---|
| | 1997 | 2007 | 2017 | 1997–2007 | 2007–2017 | 1997–2017 |
| Ganzi | 59.99 | 46.13 | 198.56 | −1.39↓ | 15.24↑ | 6.93↑ |
| Seda | 94.17 | 112.35 | 301.16 | 1.82↑ | 18.88↑ | 10.35↑ |
| Shiqu | 183.52 | 351.27 | 399.04 | 16.78↑ | 4.78↑ | 10.78↑ |
| Luhuo | 419.96 | 373.54 | 668.15 | −4.64↓ | 29.46↑ | 12.41↑ |
| Luding | 508.59 | 773.07 | 676.61 | 26.45↑ | −9.65↓ | 8.40↑ |
| Dege | 563.98 | 485.75 | 1065.39 | −7.82↓ | 57.96↑ | 25.07↑ |
| Daocheng | 1043.34 | 1390.03 | 1946.34 | 34.67↑ | 55.63↑ | 45.15↑ |
| Derong | 1048.82 | 910.59 | 1251.41 | −13.82↓ | 34.08↑ | 10.13↑ |
| Xinlong | 1125.43 | 1060.24 | 1428.20 | −6.52↓ | 36.80↑ | 15.14↑ |
| Daofu | 1205.78 | 1231.55 | 1627.67 | 2.58↑ | 39.61↑ | 21.09↑ |
| Baiyu | 1607.75 | 1675.89 | 1907.32 | 6.81↑ | 23.14↑ | 14.98↑ |
| Batang | 1631.04 | 1590.40 | 2007.33 | −4.06↓ | 41.69↑ | 18.81↑ |
| Litang | 1675.47 | 1781.81 | 2178.15 | 10.63↑ | 39.63↑ | 25.13↑ |
| Xiangcheng | 1791.34 | 1847.83 | 2239.37 | 5.65↑ | 39.15↑ | 22.40↑ |
| Danba | 1909.69 | 1914.46 | 2107.54 | 0.48↑ | 19.31↑ | 9.89↑ |
| Jiulong | 2011.38 | 2130.04 | 2551.53 | 11.87↑ | 42.15↑ | 27.01↑ |
| Kangding | 2194.07 | 2446.18 | 2542.86 | 25.21↑ | 9.67↑ | 17.44↑ |
| Yajiang | 2970.32 | 3084.52 | 3678.01 | 11.42↑ | 59.35↑ | 35.38↑ |
| Total | 22,044.66 | 23,205.65 | 28,774.61 | 116.10↑ | 556.90↑ | 336.50↑ |

Note: "↑" means increase, and "↓" means decrease.

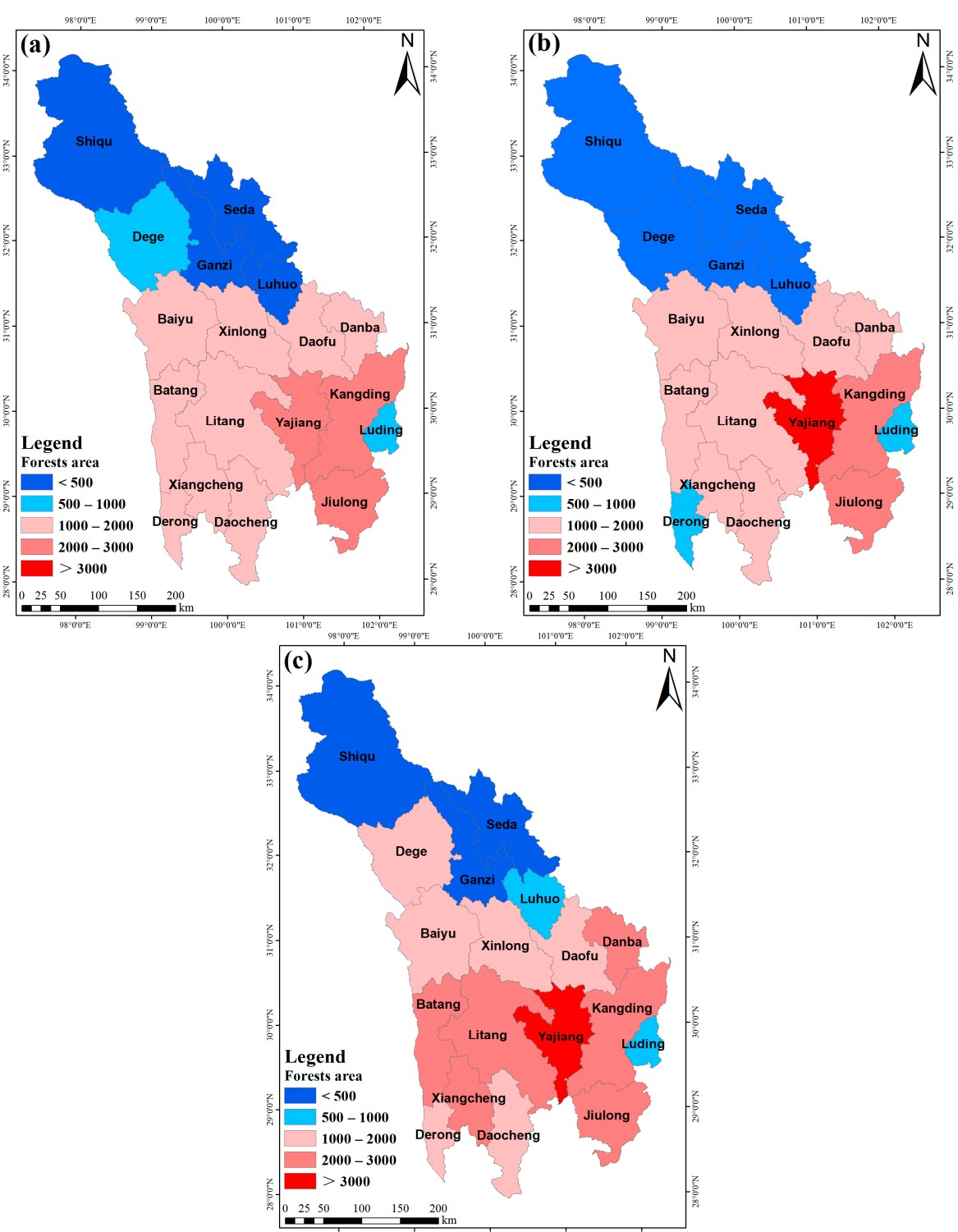

**Figure 4.** Spatial patterns of forests area of Ganzi Prefecture in different counties from 1997 to 2017 (km$^2$): (**a**) 1997, (**b**) 2007, and (**c**) 2017.

From 1997 to 2017, all counties had a growing trend in terms of the alterations to the forest area, indicating that the effects of implemented forestry ecological projects and protection policies on the increase in and protection of forests were extremely significant. The top three counties with the most significant increases were Daocheng, Yajiang, and Jiulong, recording rates of increase of 45.15 km$^2$/a, 35.38 km$^2$/a, and 27.01 km$^2$/a, respectively. The three counties with the lowest increases were Ganzi, Luding, and Danba, recording rates of increase of 6.93 km$^2$/a, 8.40 km$^2$/a, and 9.89 km$^2$/a, respectively.

From 1997 to 2007, the forest areas of Derong, Dege, Xinlong, Ganzi, Batang, and Luhuo showed decreasing trends. Among these counties, the county with the most significant decrease in forest area was Derong, recording a rate of decrease of $-13.82$ km$^2$/a; the county with the most minor drop in forest area was Ganzi, recording a rate of decrease of $-1.39$ km$^2$/a. Forests in other counties all showed increasing trends, and the county with the most significant increase in forest area was Daocheng, recording a rate of 34.67 km$^2$/a. The county with the slightest increase in forest area was Danba, recording a rate of 0.48 km$^2$/a. The results demonstrate that, at the time, Ganzi Prefecture was in the early stage of implementing forestry ecological projects and protection policies, and although forests were generally increasing in area, there were large differences in forest changes between different regions.

From 2007 to 2017, the forestry ecological projects and conservation policies implemented at th4 time had a more significant effect in Ganzi Prefecture, as only the forest area of Luding decreased, recording a decrease of $-9.65$ km$^2$/a, and forests in all other counties increased. Among these counties, the three counties with the largest increases were Yajiang, Dege, and Daocheng, recording increases of 59.35 km$^2$/a, 57.96 km$^2$/a, and 55.63 km$^2$/a, respectively; The three counties with the lowest increases in forest area were Shiqu, Kangding, and Ganzi County, recording rates of increase of 4.78 km$^2$/a, 9.67 km$^2$/a, and 15.24 km$^2$/a, respectively.

### 3.2. Changes in Forest ESV

According to Table 6, from 1997 to 2017, forest ESV in Ganzi Prefecture continued to increase, recording an overall increase of 30.22%. The improvement in the forest ESV was greatly aided by the needle-leaf and broad-leaf forests. The needle-leaf forests' ecosystem services increased by $200.18 \times 10^8$ yuan, representing a rate of increase of 31.87% compared to 1997; the ESV of broad-leaf forests increased by $59.24 \times 10^8$ yuan, recording a high rate of 26.90%. However, the mixed forests' ESV fell by $1.84 \times 10^8$ yuan, representing a drop of 45.98%.

**Table 6.** ESV changes in several forest types between 1997 and 2017.

| Forest Types | ESV ($\times 10^8$ yuan) | | | Changes Rate (%) | | |
|---|---|---|---|---|---|---|
| | 1997 | 2007 | 2017 | 1997–2007 | 2007–2017 | 1997–2017 |
| Needle-leaf forests | 628.13 | 660.78 | 828.31 | 5.2↑ | 25.35↑ | 31.87↑ |
| Broad-leaf forests | 220.27 | 234.81 | 279.51 | 6.6↑ | 19.04↑ | 26.90↑ |
| Mixed forests | 4 | 1.81 | 2.16 | −54.82↓ | 19.57↑ | −45.98↓ |
| Total | 852.39 | 897.39 | 1109.98 | 5.28↑ | 23.69↑ | 30.22↑ |

Note: "↑" means increase, and "↓" means decrease.

From the standpoint of the ESV composition of each forest type, the needle-leaf and broad-leaf forests were the forest types that provided the most values to the forest ESV composition, making up more than 99% of all forest system values (Figure 5).

The execution of forestry ecological projects and thorough ecological management have supported the restoration of forests and facilitated an increase in the forest ESV. As seen in Table 7 and Figure 6, between 1997 and 2017, climate regulation, hydrological regulation, and soil conservation contributed the most values to the forest ESV, having contributions of 70.59%, 70.60%, and 70.59%, respectively, and the values of all individual ecosystem services functions continue to grow, with climate regulation increasing to the greatest extent, while nutrient cycling increases to the least extent, and the growth rate from 1997 to 2007 was lower than the growth rate from 2007 to 2017.

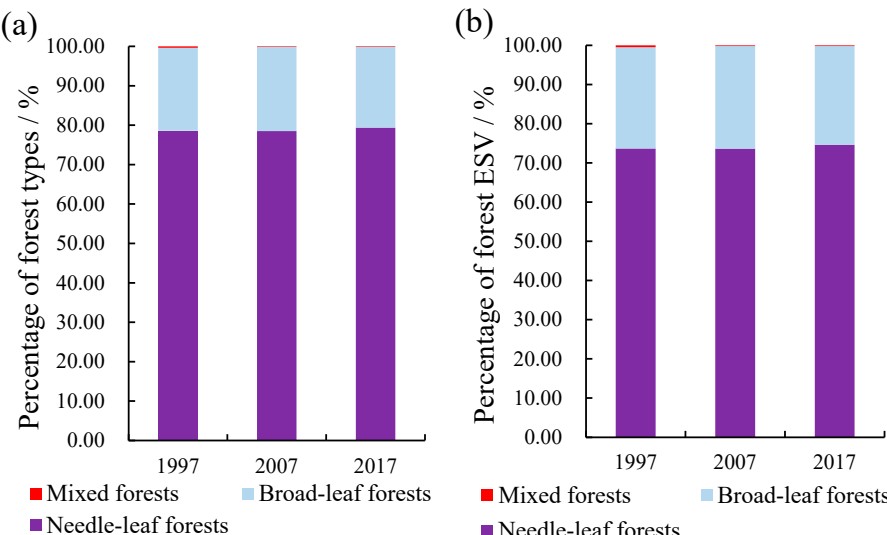

**Figure 5.** The ESV of various forest types and the percentage of each forest type: (**a**) % of different types of forest area; (**b**) % of different types of forest ESV.

**Table 7.** Forest ESV components change in Ganzi Prefecture between 1997 and 2017.

| Ecosystem Service | ESV ($\times 10^8$ yuan) | | | Changes Rate ($\times 10^8$ yuan/a) | | |
|---|---|---|---|---|---|---|
| | **1997** | **2007** | **2017** | **1997–2007** | **2007–2017** | **1997–2017** |
| Food production | 10.72 | 11.28 | 13.96 | 0.06↑ | 0.27↑ | 0.16↑ |
| Raw material | 25.09 | 26.41 | 32.68 | 0.13↑ | 0.63↑ | 0.38↑ |
| Water supply | 13.00 | 13.68 | 16.93 | 0.07↑ | 0.33↑ | 0.20↑ |
| Gas regulation | 82.15 | 86.47 | 106.98 | 0.43↑ | 2.05↑ | 1.24↑ |
| Climate regulation | 245.27 | 258.17 | 319.39 | 1.29↑ | 6.13↑ | 3.71↑ |
| Purify environment | 72.26 | 76.07 | 94.10 | 0.38↑ | 1.80↑ | 1.09↑ |
| Hydrological regulation | 165.78 | 174.67 | 215.88 | 0.89↑ | 4.12↑ | 2.50↑ |
| Soil conservation | 99.74 | 104.99 | 129.88 | 0.52↑ | 2.49↑ | 1.51↑ |
| Nutrient cycle | 7.69 | 8.09 | 10.02 | 0.04↑ | 0.19↑ | 0.12↑ |
| Biodiversity | 90.94 | 95.73 | 118.43 | 0.48↑ | 2.27↑ | 1.37↑ |
| Aesthetic landscape | 39.75 | 41.84 | 51.76 | 0.21↑ | 0.99↑ | 0.60↑ |
| Total | 852.39 | 897.39 | 1109.98 | 4.49↑ | 21.27↑ | 12.88↑ |

Note: "↑" means increase, and "↓" means decrease.

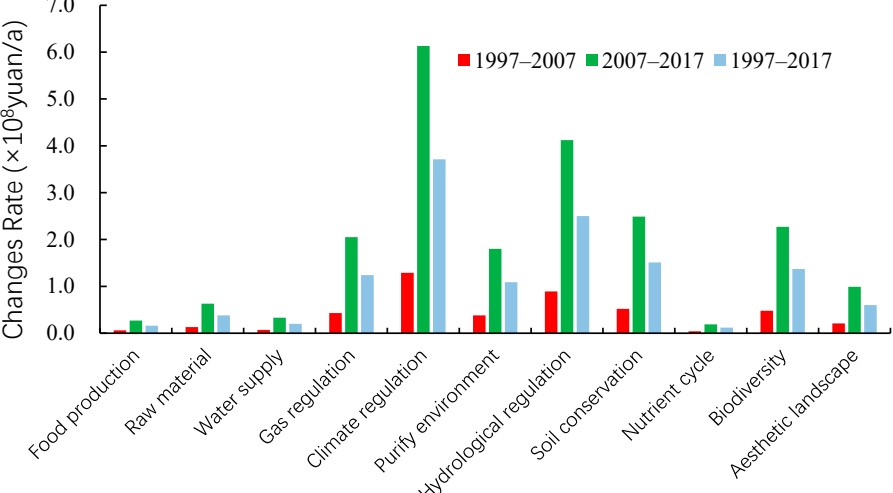

**Figure 6.** The rate of change in Ganzi Prefecture's forest ecosystem service functions between 1997 and 2017.

The factors outlined above were ranked based on the overall value of the forest service functions as follows: nutrient cycle < food production < water supply < raw material < aesthetic landscape < purify environment < gas regulation < biodiversity < soil conservation < hydrology regulation < climate regulation (Figure 7).

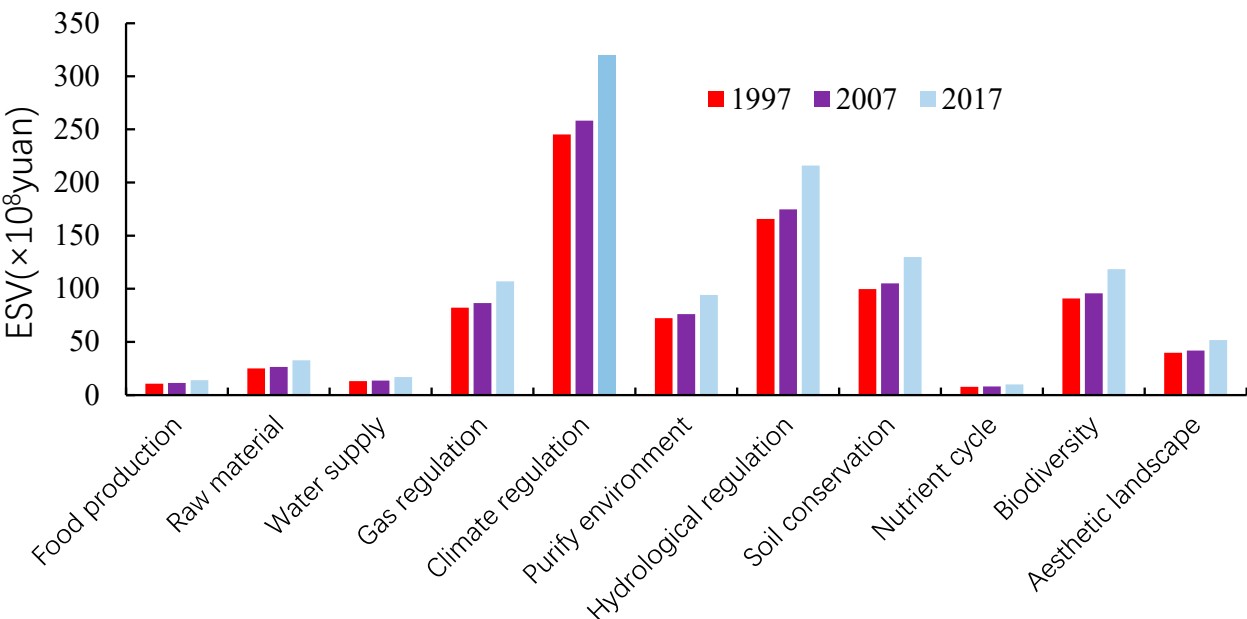

**Figure 7.** Ganzi Prefecture's forest ecosystem services and value mix from 1997 to 2017.

There were significant differences in the spatial distributions of forest ESVs in Ganzi Prefecture (Table 8). The highest forest ESV area was mainly located in Yajiang, where forest ESV exceeds 100 billion yuan. The lowest forest ESV area was mainly located in Ganzi, which forest ESV was less than 10 billion yuan (Figure 8).

**Table 8.** Forest ESV changes in various counties between 1997 and 2017.

| Forest | ESV ($\times 10^8$ yuan) | | | Changes Rate ($\times 10^8$ yuan/a) | | |
|---|---|---|---|---|---|---|
| | **1997** | **2007** | **2017** | **1997–2007** | **2007–2017** | **1997–2017** |
| Ganzi | 2.18 | 1.67 | 7.20 | −0.05↓ | 0.55↑ | 0.25↑ |
| Seda | 3.43 | 4.10 | 10.99 | 0.07↑ | 0.69↑ | 0.38↑ |
| Shiqu | 6.65 | 12.74 | 14.47 | 0.61↑ | 0.17↑ | 0.39↑ |
| Luhuo | 15.45 | 13.79 | 24.59 | −0.17↓ | 1.08↑ | 0.46↑ |
| Luding | 18.85 | 29.21 | 25.24 | 1.04↑ | −0.40↓ | 0.32↑ |
| Dege | 20.98 | 18.23 | 39.26 | −0.28↓ | 2.10↑ | 0.91↑ |
| Daocheng | 39.97 | 53.43 | 74.64 | 1.35↑ | 2.12↑ | 1.73↑ |
| Derong | 38.44 | 34.23 | 46.35 | −0.42↓ | 1.21↑ | 0.40↑ |
| Xinlong | 42.08 | 39.51 | 53.56 | −0.26↓ | 1.40↑ | 0.57↑ |
| Daofu | 45.83 | 46.45 | 62.17 | 0.06↑ | 1.57↑ | 0.82↑ |
| Baiyu | 66.18 | 68.35 | 77.99 | 0.22↑ | 0.96↑ | 0.59↑ |
| Batang | 64.89 | 62.48 | 79.78 | −0.24↓ | 1.73↑ | 0.74↑ |
| Litang | 62.27 | 66.10 | 80.67 | 0.38↑ | 1.46↑ | 0.92↑ |
| Xiangcheng | 67.43 | 69.52 | 84.89 | 0.21↑ | 1.54↑ | 0.87↑ |
| Danba | 75.99 | 76.52 | 83.75 | 0.05↑ | 0.72↑ | 0.39↑ |
| Jiulong | 78.71 | 82.64 | 99.01 | 0.39↑ | 1.64↑ | 1.01↑ |
| Kangding | 86.23 | 97.10 | 101.11 | 1.09↑ | 0.40↑ | 0.74↑ |
| Yajiang | 116.83 | 121.32 | 144.32 | 0.45↑ | 2.30↑ | 1.37↑ |
| Total | 852.39 | 897.39 | 1109.98 | 4.50↑ | 21.26↑ | 12.88↑ |

Note: "↑" means increase, and "↓" means decrease.

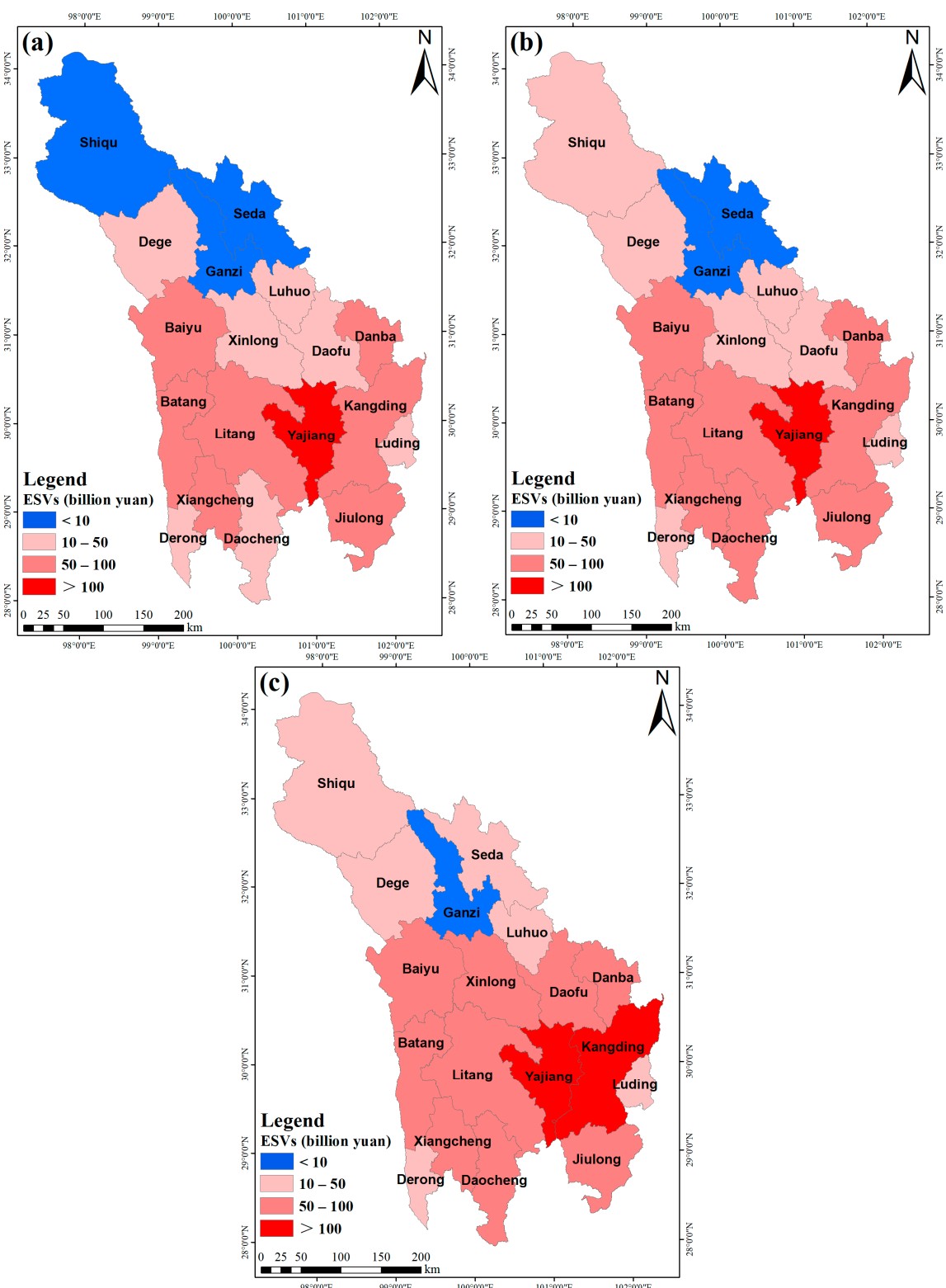

**Figure 8.** Forest ESVs' spatial patterns in various counties: (**a**) 1997, (**b**) 2007, and (**c**) 2017.

As shown in Table 7 and Figure 9, from 1997 to 2017, the forest ESV in all counties showed an increasing trend; the forest ESV of Daocheng increased at the fastest rate, i.e., $1.73 \times 10^8$ yuan/a, followed by Yajiang, which had a rate of increase of $1.37 \times 10^8$ yuan/a. However, Ganzi had the slowest rate of increase, which was only $0.25 \times 10^8$ yuan/a.

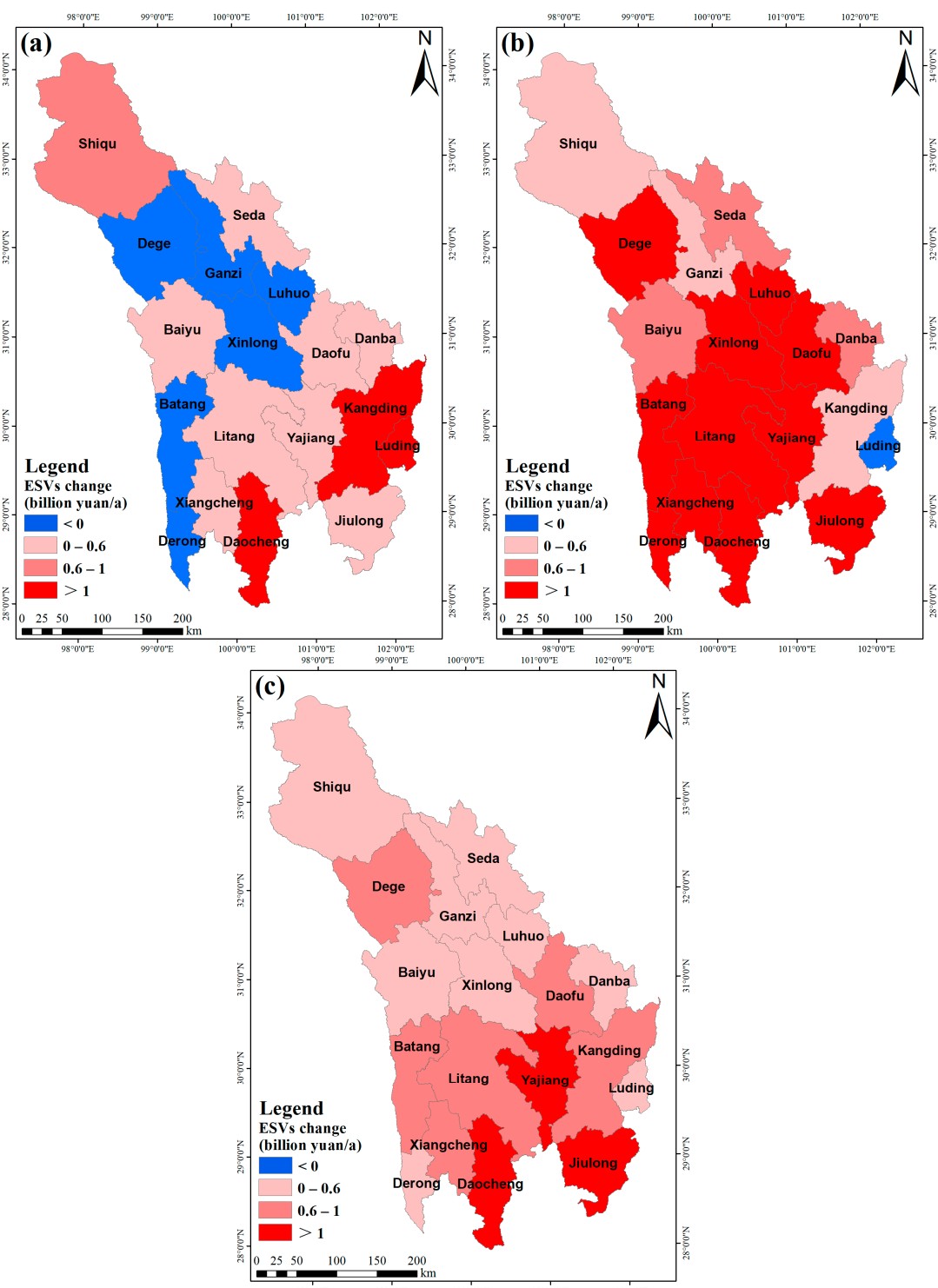

**Figure 9.** Forest ESVs' spatial pattern changes in different counties and different periods: (**a**) 1997–2007, (**b**) 2007–2017, and (**c**) 1997–2017.

As shown in Table 9 and Figure 10, in the period 1997–2017, the forest ESV density increased. The forest ecosystem service value density of Yajiang, Danba, Xiangcheng, Derong, and Jiulong was greater than $100 \times 10^4$ yuan/km², while the forest ecosystem service value density of Ganzi and Shiqu was less than $10 \times 10^4$ yuan/km².

**Table 9.** Changes in forest ESV density values in different counties from 1997 to 2017.

| Forest | ESV density (×10⁴ yuan/km²) | | | Changes Rate (×10⁴ yuan/km²) | | |
|---|---|---|---|---|---|---|
| | 1997 | 2007 | 2017 | 1997–2007 | 2007–2017 | 1997–2017 |
| Ganzi | 2.98 | 2.29 | 9.86 | −0.69↓ | 7.57↑ | 6.88↑ |
| Seda | 3.67 | 4.39 | 11.77 | 0.72↑ | 7.38↑ | 8.10↑ |
| Shiqu | 2.67 | 5.11 | 5.80 | 2.44↑ | 0.69↑ | 3.13↑ |
| Luhuo | 33.58 | 29.98 | 53.43 | −3.60↓ | 23.46↑ | 19.86↑ |
| Luding | 87.07 | 134.94 | 116.58 | 47.87↑ | −18.36↓ | 29.51↑ |
| Dege | 19.03 | 16.54 | 35.61 | −2.50↓ | 19.08↑ | 16.58↑ |
| Daocheng | 54.58 | 72.97 | 101.92 | 18.39↑ | 28.95↑ | 47.34↑ |
| Derong | 131.83 | 117.39 | 158.94 | −14.44↓ | 41.56↑ | 27.11↑ |
| Xinlong | 49.10 | 46.10 | 62.50 | −3.00↓ | 16.39↑ | 13.39↑ |
| Daofu | 64.98 | 65.86 | 88.15 | 0.88↑ | 22.28↑ | 23.17↑ |
| Baiyu | 63.72 | 65.81 | 75.09 | 2.09↑ | 9.28↑ | 11.37↑ |
| Batang | 82.65 | 79.57 | 101.61 | −3.08↓ | 22.04↑ | 18.96↑ |
| Litang | 44.49 | 47.22 | 57.63 | 2.74↑ | 10.41↑ | 13.15↑ |
| Xiangcheng | 134.43 | 138.59 | 169.24 | 4.16↑ | 30.64↑ | 34.80↑ |
| Danba | 163.20 | 164.35 | 179.87 | 1.14↑ | 15.53↑ | 16.67↑ |
| Jiulong | 116.33 | 122.14 | 146.33 | 5.80↑ | 24.20↑ | 30.00↑ |
| Kangding | 75.07 | 84.54 | 88.03 | 9.47↑ | 3.49↑ | 12.96↑ |
| Yajiang | 154.58 | 160.51 | 190.95 | 5.93↑ | 30.44↑ | 36.37↑ |
| Total | 55.71 | 58.65 | 72.55 | 2.94↑ | 13.89↑ | 16.84↑ |

Note: "↑" means increase, and "↓" means decrease.

As shown in Table 9 and Figure 11, in the period 1997–2017, the forest ESV density of all counties demonstrated a rising trend; among these counties, the forest ESV density of Daocheng County increased to the greatest extent, recording an increase of $47.34 \times 10^4$ yuan/km²; however, the forest ESV density of Ganzi County increased to the least extent, as the forest ESV density increase was only $6.88 \times 10^4$ yuan/km².

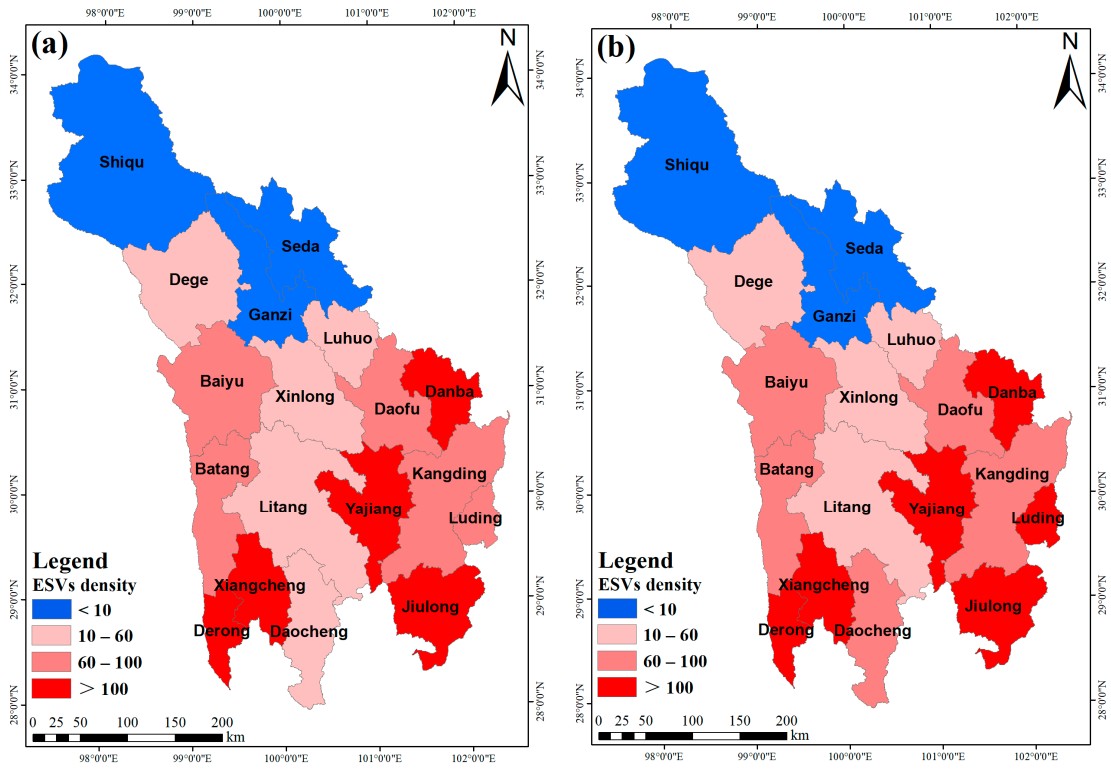

**Figure 10.** *Cont.*

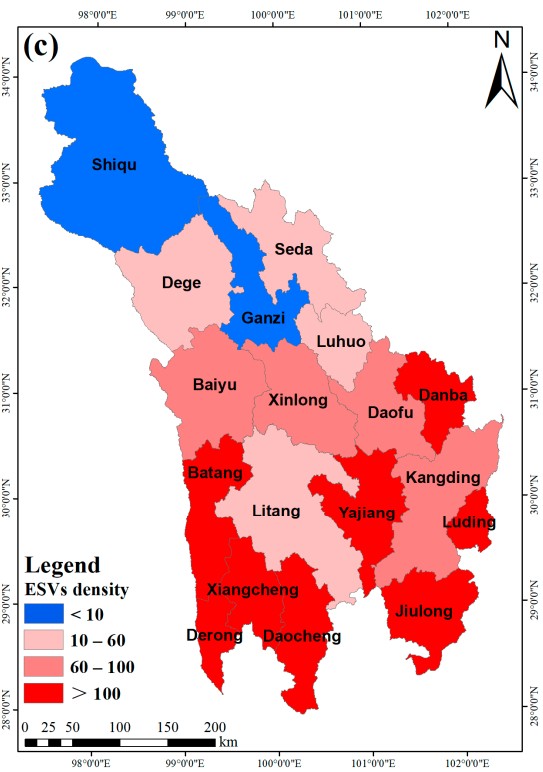

**Figure 10.** Forest ESVs' density spatial patterns in different counties ($\times 10^4/\mathrm{km}^2$). (**a**) 1997, (**b**) 2007, and (**c**) 2017.

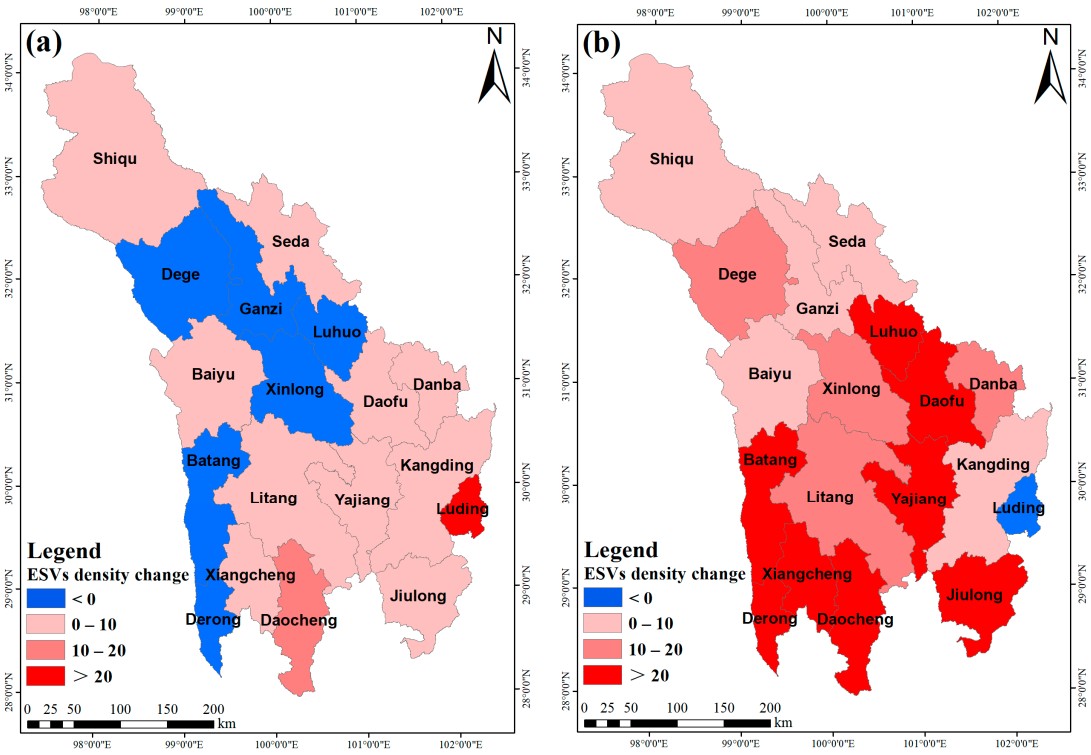

**Figure 11.** *Cont.*

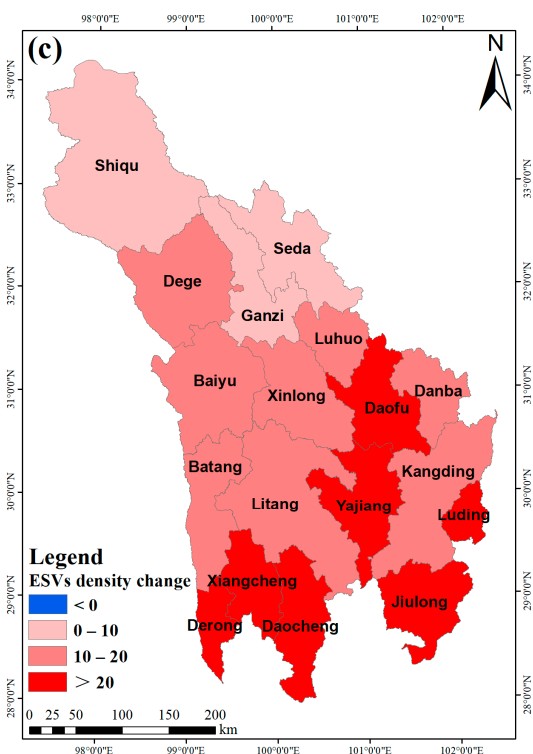

**Figure 11.** Forest ESVs' density spatial pattern changes in different counties ($\times 10^4$/km$^2$) and different periods: (**a**) 1997–2007, (**b**) 2007–2017, and (**c**) 1997–2017.

## 4. Discussion

### 4.1. Effects of Forest Area Change on ESV

Forest ESV is directly affected by changes in forest areas. Over the past century, China's forests have undergone a significant transformation, mostly occurring as a result of the implementation of initiatives to encourage forest conservation and restoration [46]. According to the 2022 bulletin on the state of China's land greening, China's forests cover 231 million hectares, and its forest coverage rate was 24.02%. At the moment, China ranks fifth in the world in terms of the area of forest resources, and it ranks first in terms of the area of planted forests. Similarly, the forests of Ganzi Prefecture have benefited from the implementation of China's forest protection and restoration program and undergone dramatic changes. The high altitude and cold climate limit the development of agriculture in Ganzi Prefecture. Forestry was, therefore, Ganzi Prefecture's primary economic driver before 1997, but excessive deforestation has caused the forest's ecosystem services to drastically decline, which has had a significant negative impact on the area's ecological environment. The region has experienced forest degradation, biodiversity loss, and accelerated soil erosion, as well as an increase in geologic hazards (landslides, avalanches, mudslides, earthquakes), all of which have seriously harmed human life and impeded the area's ability to achieve sustainable development. In order to improve the regional ecological environment and address the issue of forest degradation, Ganzi Prefecture implemented key forestry ecological projects and protection policies, such as the natural forest protection project, the ban on logging in natural forests, the return of farmland to forests and grasslands project, the protection of wild fauna and flora, and the construction of nature reserves, one after another, in the period after 1998. Through this study, it was found that the implementation of forestry ecological projects and protection policies increased the forest area of Ganzi Prefecture by 6729.95 km$^2$ and the forest ESV by 257.59 $\times$ 10$^8$ yuan during the period 1997–2017, indicating that the implementation of forestry ecological projects and protection policies was crucial to increasing the forest area and the forest ESV, as well as improving the ecological environment.

As a link between the natural world and human well-being, ecosystem services are essential for the wise exploitation of natural resources [47]. With the increase in forests, the forest ESV progressively increases, and the forestry management mode in Ganzi Prefecture has changed, with forest logging replaced by forest protection, forest tourism, the development of forest products, and other forms of economic activities, which not only make the ecological environment in Ganzi Prefecture gradually change for the better, but also promote regional economic development. At present, Ganzi Prefecture has maintained a "double growth" in forest area and forest reserves for 21 consecutive years, with the total amount allowing the county to rank first in Sichuan Province, China, and the public's comprehensive satisfaction with the ecological environment has ranked first in Sichuan Province, China, for two consecutive years.

Having accurate information on forest areas is of great help to decision makers when they are making policy and investment decisions [48]. This study also demonstrated that forest ecosystem services are closely related to forest management [49] and proper forest management is crucial for increasing forest area and improving forest ESV. Therefore, in order for humans to obtain greater ecological benefits from forests, it is recommended that they implement ecological projects in a rational manner, optimize forest structures, monitor changes in forest dynamics, and continue to protect forest resources.

In this study, forest information from the period 1997–2017 was extracted using the remote sensing visual interpretation method, which provides higher precision and accuracy of extracted forest data relative to computerized extraction. Based on the extracted forest information, this study utilized the adjusted local value coefficients to quantitatively assess the forest ESV of Ganzi Prefecture in time and space, and compared to other forest ESV assessment methods, the method used in this study has the advantage of being relatively simple, requiring fewer indicators, conforming to the actual situation of the region, and rapidly quantifying the forests' ESVs, meaning that the ecological value of the forests can be quickly grasped. This study has special theoretical and practical significance in terms of grasping the dynamics of forest resources and protecting and restoring forest ecosystems. The intuitive monetary value highlights how crucial forest resources are to both the local economy and the ecosystem. The results of this study give people a better understanding of the value of forest ecosystem services in Ganzi Prefecture while also serving as a scientific guide for their optimal management, protection, and sustainable usage. Additionally, it offers a theoretical scientific foundation for the protection and management of forest ecosystems around the world.

### 4.2. Suggestions

Forests create important habitats for biodiversity conservation and supply a variety of vital ecosystem services that are of great importance to human society, humans increasingly recognize the significance of forest ecosystem services [50], and there is growing evidence that biodiversity contributes to forest ecosystem's functions and service supply [51–53]. Conservation management is also increasingly emphasizing the co-conservation or improvement of ecosystem services and biodiversity [54]. Therefore, it is recommended that forest managers give more consideration to planting mixed forests when planting plantations, mainly because mixed forests contribute to increased forest diversity and will help forests to provide a broader range of ecosystem services. As a result, the following suggestions are particularly important to this study:

(1) Protecting natural forests is more important than afforestation [55]; thus, it is recommended to focus on protecting these forests, continue to implement the national forest protection policy, and prohibit the logging of natural forests.

(2) In the future, attention must be paid to the cultivation and tending of mixed forests in order to improve the biodiversity of forests.

(3) Protect existing forest resources and give full attention to forests' advantages, strengthen the prevention and management of forest diseases and insect pests, and protect against forest fires.

(4)　Focus on regions in which the forest area and forest ESVs have decreased.

*4.3. Future Work*

This study has significant implications for the sustainable development of forests, the protection of forests at a regional scale, the later formulation of forest protection strategies in ecologically fragile areas, and the preservation of local ecological security. It has great significance in terms of the preservation of the forest environment and the development of the southeastern margin of the Tibetan Plateau. At the same time, this research offers a case study of forest ESVs' spatio-temporal dynamics, which possess particular value as a reference for the formulation of forest protection policies. The following issues are the study's shortcomings: (1) The spatial resolution of the remote sensing data had an impact on the categorization of forests. (2) Moreover, the effects of height on forest ESVs are disregarded in this study. We will, thus, investigate the effects of environmental factors, such as climate, soil, and topography, on forest area and forest ESV distribution in different regions in order to rationalize forest conservation actions.

## 5. Conclusions

This study carried out a quantitative analysis of both spatial and temporal shifts in forest area and forest ESVs in Ganzi Prefecture from 1997 to 2017 using Landsat TM/OLI satellite imagery data and GIS technology. The primary study results revealed the following conclusions: (1) From 1997 to 2017, the total forest area of Ganzi Prefecture showed an initially slow and then strong increasing trend, with a total increase of 6729.95 km$^2$ and an overall rate of change of 336.50 km$^2$/a. (2) From 1997 to 2017, the forest ESV in Ganzi Prefecture continued to increase; by 2017, forest ESV had increased by a total of $257.59 \times 10^8$ yuan from 1997, representing an overall increase of 30.22%. (3) The implementation of forestry engineering and protection policies has had a beneficial influence in terms of enhancing forest area and forest ESV in Ganzi Prefecture. (4) It is suggested that Ganzi Prefecture could strengthen forest management in areas with reduced forests; emphasize the cultivation and management of mixed forests, especially the protection of natural forests; and continue to implement key forestry-related ecological projects in the future in order to improve the ecological well-being of the local population. Through this study, we have grasped the spatial distribution characteristics of forest area and forest ESV in Ganzi Prefecture. Next, we will intensify our efforts to carry out research into the interactions between forest distribution patterns and environmental factors, providing a scientific theoretical basis for the development of forest management methods suitable for each region, and this approach will help us to realize the sustainable development of the forests in the Hengduan Mountains.

**Author Contributions:** Conceptualization, Y.W. and X.B.; software, investigation, validation, data curation, Y.W. and X.B.; writing—original draft preparation, Y.W.; writing—review and editing, Q.L. and G.W.; supervision, J.G. and P.P. All authors have read and agreed to the published version of the manuscript.

**Funding:** This research was financially supported by the Second Tibetan Plateau Scientific Expedition and Research Program (STEP), China (2019QZKK0301).

**Data Availability Statement:** The data presented in this study are available on request from the corresponding author. The data are not publicly available due to all data generated during this study are included in this article.

**Acknowledgments:** We thank the editors and reviewers for their detailed and constructive comments, and we acknowledge the geospatial data cloud, which provided freely accessible remote sensing images.

**Conflicts of Interest:** The authors declare there are no conflicts of interest.

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
