# Peer review of "Spatio-Temporal Changes in Forest Area and Its Ecosystem Service Value in Ganzi Prefecture, China, in the Period 1997–2017"

_forests, doi:10.3390/f14091731_

Round 1
Reviewer 1 Report
- The topic is critical, especially in the current context of climate change.
- The ecological benefits of forests are essential, but they should be analysed in relation to the environmental footprint of the area's economic activities. Does the increase in forested areas cover the growth of economic activities and their impact? This issue should also be assessed or, at least, discussed in the present paper. At this point, the discussions and suggestions are too general and limited.
- Also, the present approach should be contextualised in the introduction or in discussions. What is the state of forests in China? What are the general differences among different regions?
- Also, the geographic differences in the natural potential should be discussed in relation to the actual explanation of the policies, measures and actions taken within the analysed period 1997-2017 that contributed to forest dynamics.
- Some many figures and tables are not enough comments. The authors should include more details in commenting on the maps/tables or put some figures/tables as annexes.
- The existing comments on the resulting maps should also be more detailed and highlight the causes of the spatial differentiation and not just general ideas about this distribution.
- Figure 1 – please check the validity of the map. Also, the colours should be chosen differently (e.g. from green to brown)
- The differences between the three years are not visible. A map showing just the differences would be suitable. Also, in the text, the authors should comment on these modifications in relation to other land uses that replaced/were replaced by forests.
- Fig 3. The unit for forest areas (ha? sqkm?) is not stated
- Also, in the text, the authors should comment on these modifications in relation to other land uses that replaced/were replaced by forests.
- Table 7 – the intervals overlap Ex. 1997-2007, 2007-2017. 2007 is in the first or the second period? Should it be 1997-2006, 2007-2017? Or 1997-2007, 2008-2017?
Reviewer 2 Report
- This work presented a quantitative assessment for the spatio-temporal dynamic changes of forest ecosystem service value (ESV) in China from 1997 to 2017 using Landsat images time series.
- Line 16: remote sensing photos >> remote sensing images
- Line 18: what are global value coefficients and adjusted local value coefficients? Their definition?
- Line 20: 257.59 × 108 yuan? On what basis?
- Some abbreviations are repeated several times in the text. Such as ES (lines 32, 36)/ RS (lines 65, 66, 72, 75, 80). Abbreviation should be used only the first time the original word appears. Of course, remote sensing should not be used as an abbreviation.
- Line 79: The origin of the two words of UAV and LiDAR should be written in the text.
- In the introduction, the problem is not very clear. The end of the third paragraph and also, paragraph 4 were explaining about the application and importance of remote sensing without any precise target (Consistent with the topic of the article).
- The background of the research concerned on the use of remote sensing in the field of ESV may be strengthened.
- In addition, the innovation should be more prominent in the text.
- Provide a table for the properties of the Landsat data used in the paper.
- It is better to present a graphical abstract for methodology of the paper.
- The role of remote sensing images in the change process was not very clear! What is the exact role of RS in this paper? (refer to previous comment)
- How to produce forest maps of figure 2? What the using methods?
- What were the accuracies of these maps?
- Quality control or accuracy assessment is a critical stage in change detection process using RS images. This issue was a bit weak in this paper.
- Suggestions and future work may be provided in the conclusion.
- Conclusion was so weak.
Needs a moderate English editing.
Round 2
Reviewer 2 Report
Thanks for your responses. Your paper was revised very well.
However, I emphasize that the flowchart of methodology be added (at a figure). (Comment 11)
Average
